# Countering Adversarial Images using Input Transformations

**Chuan Guo**[*]
Cornell University

**Mayank Rana & Moustapha Cissé & Laurens van der Maaten**
Facebook AI Research

## Abstract

This paper investigates strategies that defend against adversarial-example attacks on image-classification systems by transforming the inputs before feeding them to the system. Specifically, we study applying image transformations such as bit-depth reduction, JPEG compression, total variance minimization, and image quilting before feeding the image to a convolutional network classifier. Our experiments on ImageNet show that total variance minimization and image quilting are very effective defenses in practice, in particular, when the network is trained on transformed images. The strength of those defenses lies in their non-differentiable nature and their inherent randomness, which makes it difficult for an adversary to circumvent the defenses. *Our best defense eliminates* 60% *of strong gray-box and* 90% *of strong black-box attacks by a variety of major attack methods.*

## 1 Introduction

As the use of machine intelligence increases in security-sensitive applications (Bojarski et al., 2016; Amodei et al., 2015), robustness has become a critical feature to guarantee the reliability of deployed machine-learning systems. Unfortunately, recent research has demonstrated that existing models are not robust to small, adversarially designed perturbations of the input (Biggio et al., 2013; Szegedy et al., 2014; Goodfellow et al., 2015; Kurakin et al., 2016a; Cisse et al., 2017a). Adversarially perturbed examples have been deployed to attack image classification services (Liu et al., 2016), speech recognition systems (Cisse et al., 2017a), and robot vision (Melis et al., 2017). The existence of these *adversarial examples* has motivated proposals for approaches that increase the robustness of learning systems to such examples (Papernot et al., 2016; Kurakin et al., 2016a; Cisse et al., 2017b).

The robustness of machine learning models to adversarial examples depends both on the properties of the model (*i.e.*, Lipschitzness) and on the nature of the problem considered, *e.g.*, on the input dimensionality and the Bayes error of the problem (Fawzi et al., 2015; 2016). Consequently, *defenses* that aim to increase robustness against adversarial examples fall in one of two main categories. The first category comprises *model-specific* strategies that enforce model properties such as invariance and smoothness via the learning algorithm or regularization scheme (Shaham et al., 2015; Kurakin et al., 2016a; Cisse et al., 2017b), potentially exploiting knowledge about the adversary's attack strategy (Goodfellow et al., 2015). The second category of defenses are *model-agnostic*: they try to remove adversarial perturbations from the input. For example, in the context of image classification, adversarial perturbations can be partly removed via JPEG compression (Dziugaite et al., 2016) or image re-scaling (Lu et al., 2017). Hitherto, none of these defenses has been shown to be very effective. Specifically, model-agnostic defenses appear too simple to sufficiently remove adversarial perturbations from input images. By contrast, model-specific defenses make strong assumptions about the nature of the adversary (*e.g.*, on the norm that the adversary minimizes or on the number of iterations it uses to generate the perturbation). Consequently, they do not satisfy Kerckhoffs (1883) principle: the adversary can alter its attack to circumvent such model-specific defenses.

In this paper, we focus on increasing the effectiveness of model-agnostic defense strategies by developing approaches that (1) remove the adversarial perturbations from input images, (2) maintain sufficient information in input images to correctly classify them, and (3) are still effective in settings in which the adversary has information on the defense strategy being used. We explore transformations based on image cropping and rescaling (Graese et al., 2016), bit-depth reduction (Xu et al.,

---

[*]This work was performed whilst Chuan Guo was at Facebook AI Research.

2017), JPEG compression (Dziugaite et al., 2016), total variance minimization (Rudin et al., 1992), and image quilting (Efros & Freeman, 2001). We show that these defenses can be surprisingly effective against existing attacks, in particular, when the convolutional network is trained on images that are transformed in a similar way. The image transformations are good at countering the (iterative) fast gradient sign method (Kurakin et al., 2016a), Deepfool (Moosavi-Dezfooli et al., 2016), and the Carlini & Wagner (2017) attack, even in gray-box settings in which the model architecture and parameters are public. Our strongest defenses are based on total variation minimization and image quilting: these defenses are non-differentiable and inherently random, which makes it difficult for an adversary to get around them. *Our best defenses eliminate* $60\%$ *of gray-box attacks and* $90\%$ *of black-box attacks by four major attack methods that perturb pixel values by* $8\%$ *on average.*

## 2 PROBLEM DEFINITION

We study defenses against non-targeted adversarial examples for image-recognition systems. Let $\mathcal{X} = [0, 1]^{H \times W \times C}$ be the image space. Given an image classifier $h(\cdot)$ and a source image $\mathbf{x} \in \mathcal{X}$, a *non-targeted*[1] *adversarial example* of $\mathbf{x}$ is a perturbed image $\mathbf{x}' \in \mathcal{X}$ such that $h(\mathbf{x}) \neq h(\mathbf{x}')$ and $d(\mathbf{x}, \mathbf{x}') \leq \rho$ for some dissimilarity function $d(\cdot, \cdot)$ and $\rho \geq 0$. Ideally, $d(\cdot, \cdot)$ measures the perceptual difference between $\mathbf{x}$ and $\mathbf{x}'$ but, in practice, the Euclidean distance $d(\mathbf{x}, \mathbf{x}') = \|\mathbf{x} - \mathbf{x}'\|_2$ or the Chebyshev distance $d(\mathbf{x}, \mathbf{x}') = \|\mathbf{x} - \mathbf{x}'\|_\infty$ is most commonly used.

Given a set of $N$ images $\{\mathbf{x}_1, \ldots, \mathbf{x}_N\}$ and a target classifier $h(\cdot)$, an *adversarial attack* aims to generate $\{\mathbf{x}'_1, \ldots, \mathbf{x}'_N\}$ such that each $\mathbf{x}'_n$ is an adversarial example for $\mathbf{x}_n$. The *success rate* of an attack is measured by the proportion of predictions that was altered by an attack: $\frac{1}{N} \sum_{n=1}^{N} \mathbb{1}[h(\mathbf{x}_n) \neq h(\mathbf{x}'_n)]$. The success rate is generally measured as a function of the magnitude of the perturbations performed by the attack, using the *normalized $L_2$-dissimilarity*:

$$\frac{1}{N} \sum_{n=1}^{N} \frac{\|\mathbf{x}_n - \mathbf{x}'_n\|_2}{\|\mathbf{x}_n\|_2}. \tag{1}$$

A strong adversarial attack has a high success rate whilst its normalized $L_2$-dissimilarity is low.

In most practical settings, an adversary does not have direct access to the model $h(\cdot)$ and has to do a *black-box* attack. However, prior work has shown successful attacks by transferring adversarial examples generated for a separately-trained model to an unknown target model (Liu et al., 2016). Therefore, we investigate both the black-box and a more difficult *gray-box* attack setting: in our gray-box setting, the adversary has access to the model architecture and the model parameters, but is unaware of the defense strategy that is being used.

A *defense* is an approach that aims make the prediction on an adversarial example $h(\mathbf{x}')$ equal to the prediction on the corresponding clean example $h(\mathbf{x})$. In this study, we focus on *image-transformation defenses* $g(\mathbf{x})$ that perform prediction via $h(g(\mathbf{x}'))$. Ideally, $g(\cdot)$ is a complex, non-differentiable, and potentially stochastic function: this makes it difficult for an adversary to attack the prediction model $h(g(\mathbf{x}))$ even when the adversary knows both $h(\cdot)$ and $g(\cdot)$.

## 3 ADVERSARIAL ATTACKS

One of the first successful attack methods is the **fast gradient sign method** (FGSM; Goodfellow et al. (2015)). Let $\ell(\cdot, \cdot)$ be the differentiable loss function that was used to train the classifier $h(\cdot)$, *e.g.*, the cross-entropy loss. The FGSM adversarial example corresponding to a source input $\mathbf{x}$ and true label $y$ is:

$$\mathbf{x}' = \mathbf{x} + \epsilon \cdot \text{sign} \left( \nabla_{\mathbf{x}} \ell(\mathbf{x}, y) \right), \tag{2}$$

for some $\epsilon > 0$ that governs the perturbation magnitude. A stronger variant of this attack, called **iterative FGSM** (I-FGSM; Kurakin et al. (2016b)), iteratively applies the FGSM update:

$$\mathbf{x}^{(m)} = \mathbf{x}^{(m-1)} + \epsilon \cdot \text{sign} \left( \nabla_{\mathbf{x}^{(m-1)}} \ell(\mathbf{x}^{(m-1)}, y) \right), \tag{3}$$

---

[1]Given a target class $c$, a *targeted adversarial example* $\mathbf{x}'$ is an example that satisfies $h(\mathbf{x}') = c$. We do not consider targeted attacks in this study.

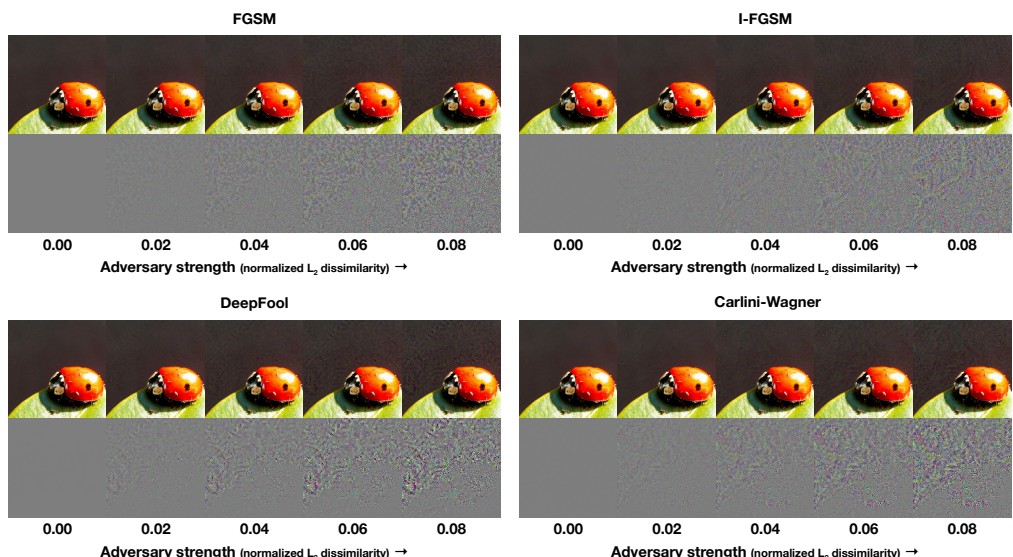

Figure 1: Adversarial images and corresponding perturbations at five levels of normalized $L_2$-dissimilarity for all four attacks.

where $m = 1, \ldots, M$; $\mathbf{x}^{(0)} = \mathbf{x}$; and $\mathbf{x}' = \mathbf{x}^{(M)}$. The number of iterations $M$ is set such that $h(\mathbf{x}') \neq h(\mathbf{x})$. Both FGSM and I-FGSM approximately minimize the Chebyshev distance between the inputs and the adversarial examples they generate.

Alternative attacks aim to minimize the Euclidean distance between the input and the adversarial example instead. For instance, assuming $h(\cdot)$ is a binary classifier, **DeepFool** (Moosavi-Dezfooli et al., 2016) projects $\mathbf{x}$ onto a linearization of the decision boundary defined by $h(\cdot)$ for $M$ iterations:

$$\mathbf{x}^{(m)} = \mathbf{x}^{(m-1)} - \epsilon \cdot \frac{h(\mathbf{x}^{(m-1)})}{\|\nabla_{\mathbf{x}^{(m-1)}} h(\mathbf{x}^{(m-1)})\|_2^2} \nabla_{\mathbf{x}^{(m-1)}} h\left(\mathbf{x}^{(m-1)}\right), \tag{4}$$

where $\mathbf{x}^{(0)}$ and $\mathbf{x}'$ are defined as in I-FGSM. The multi-class variant of DeepFool performs the projection onto the nearest class boundaries. The linearization performed in DeepFool is particularly well suited for ReLU-networks, as these represent piecewise linear class boundaries.

**Carlini-Wagner's** $L_2$ **attack** (CW-L2; Carlini & Wagner (2017)) is an optimization-based attack that combines a differentiable surrogate for the model's classification accuracy with an $L_2$-penalty term. Let $Z(\mathbf{x})$ be the operation that computes the logit vector (*i.e.*, the output before the softmax layer) for an input $\mathbf{x}$, and $Z(\mathbf{x})_k$ be the logit value corresponding to class $k$. The untargeted variant of CW-L2 finds a solution to the unconstrained optimization problem

$$\min_{\mathbf{x}'} \left[ \|\mathbf{x} - \mathbf{x}'\|_2^2 + \lambda_f \max\left(-\kappa, Z(\mathbf{x}')_{h(\mathbf{x})} - \max\{Z(\mathbf{x}')_k : k \neq h(\mathbf{x})\}\right) \right], \tag{5}$$

where $\kappa$ denotes a margin parameter, and where the parameter $\lambda_f$ trades off the perturbation norm and the hinge loss of predicting a different class. We perform the minimization over $\mathbf{x}'$ using the Adam optimizer (Kingma & Ba, 2014) for 100 iterations with an initial learning rate of 0.001.

All of the aforementioned attacks enforce that $\mathbf{x}' \in \mathcal{X}$ by clipping values between 0 and 1. Figure 1 shows adversarial images produced by all four attacks at five normalized $L_2$-dissimilarity levels.

## 4 DEFENSES

Adversarial attacks alter particular statistics of the input image in order to change the model prediction. Indeed, adversarial perturbations $\mathbf{x} - \mathbf{x}'$ have a particular structure, as illustrated by Figure 1. We design and experiment with image transformations that alter the structure of these perturbations, and investigate whether the alterations undo the effects of the adversarial attack. We investigate five image transformations: (1) image cropping and rescaling, (2) bit-depth reduction, (3) JPEG compression, (4) total variance minimization, and (5) image quilting.

## 4.1 IMAGE CROPPING-RESCALING, BIT-DEPTH REDUCTION, AND COMPRESSION

We first introduce three simple image transformations: image cropping-rescaling (Graese et al., 2016), bit-depth reduction (Xu et al., 2017), and JPEG compression and decompression (Dziugaite et al., 2016). *Image cropping-rescaling* has the effect of altering the spatial positioning of the adversarial perturbation, which is important in making attacks successful. Following He et al. (2016), we crop and rescale images at training time as part of the data augmentation. At test time, we average predictions over random image crops. *Bit-depth reduction* (Xu et al., 2017) perform a simple type of quantization that can removes small (adversarial) variations in pixel values from an image; we reduce images to 3 bits in our experiments. *JPEG compression* (Dziugaite et al., 2016) removes small perturbations in a similar way; we perform compression at quality level 75 (out of 100).

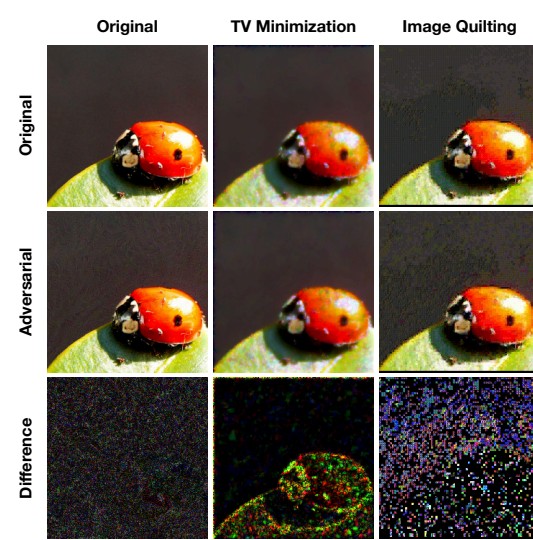

Figure 2: Illustration of total variance minimization and image quilting applied to an original and an adversarial image (produced using I-FGSM with $\epsilon = 0.03$, corresponding to a normalized $L_2$-dissimilarity of 0.075). From left to right, the columns correspond to: (1) no transformation, (2) total variance minimization, and (3) image quilting. From top to bottom, rows correspond to: (1) the original image, (2) the corresponding adversarial image produced by I-FGSM, and (3) the absolute difference between the two images above. Difference images were multiplied by a constant scaling factor to increase visibility.

## 4.2 TOTAL VARIANCE MINIMIZATION

An alternative way of removing adversarial perturbations is via a compressed sensing approach that combines pixel dropout with total variation minimization (Rudin et al., 1992). This approach randomly selects a small set of pixels, and reconstructs the "simplest" image that is consistent with the selected pixels. The reconstructed image does not contain the adversarial perturbations because these perturbations tend to be small and localized.

Specifically, we first select a random set of pixels by sampling a Bernoulli random variable $X(i, j, k)$ for each pixel location $(i, j, k)$; we maintain a pixel when $X(i, j, k) = 1$. Next, we use total variation minimization to constructs an image $\mathbf{z}$ that is similar to the (perturbed) input image $\mathbf{x}$ for the selected set of pixels, whilst also being "simple" in terms of total variation by solving:

$$\min_{\mathbf{z}} \|(1 - X) \odot (\mathbf{z} - \mathbf{x})\|_2 + \lambda_{\text{TV}} \cdot \text{TV}_p(\mathbf{z}). \tag{6}$$

Herein, $\odot$ denotes element-wise multiplication, and $\text{TV}_p(\mathbf{z})$ represents the $L_p$-total variation of $\mathbf{z}$:

$$\text{TV}_p(\mathbf{z}) = \sum_{k=1}^{K} \left[ \sum_{i=2}^{N} \|\mathbf{z}(i, :, k) - \mathbf{z}(i-1, :, k)\|_p + \sum_{j=2}^{N} \|\mathbf{z}(:, j, k) - \mathbf{z}(:, j-1, k)\|_p \right]. \tag{7}$$

The total variation (TV) measures the amount of fine-scale variation in the image $\mathbf{z}$, as a result of which TV minimization encourages removal of small (adversarial) perturbations in the image. The objective function (6) is convex in $\mathbf{z}$, which makes solving for $\mathbf{z}$ straightforward. In our implementation, we set $p = 2$ and employ a special-purpose solver based on the split Bregman method (Goldstein & Osher, 2009) to perform total variance minimization efficiently.

The effectiveness of TV minimization is illustrated by the images in the middle column of Figure 2: in particular, note that the adversarial perturbations that were present in the background for the non-transformed image (see bottom-left image) have nearly completely disappeared in the TV-minimized adversarial image (bottom-center image). As expected, TV minimization also changes image structure in non-homogeneous regions of the image, but as these perturbations were not adversarially designed we expect the negative effect of these changes to be limited.

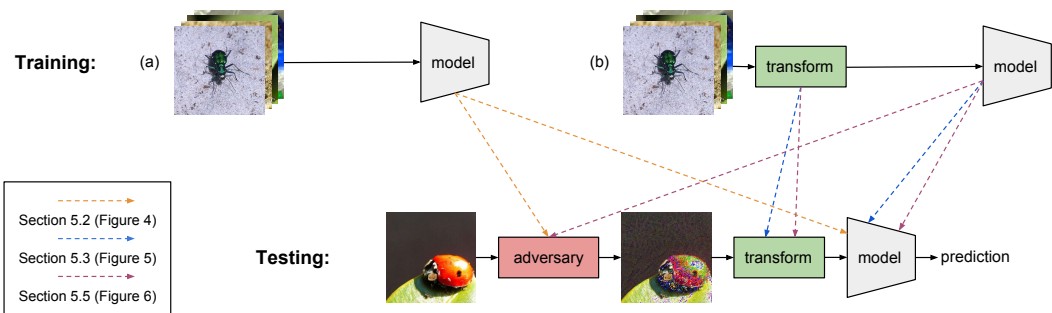

Figure 3: Block diagram detailing the differences between the experimental setups in Section 5.2, 5.3, and 5.4. We train networks (a) on regular images or (b) on transformed images; we test the networks on transformed adversarial images. For each of the three setups, dashed arrows indicate which model is used by the adversary and which model is used by the classification model.

### 4.3 IMAGE QUILTING

Image quilting (Efros & Freeman, 2001) is a non-parametric technique that synthesizes images by piecing together small patches that are taken from a database of image patches. The algorithm places appropriate patches in the database for a predefined set of grid points, and computes minimum graph cuts (Boykov et al., 2001) in all overlapping boundary regions to remove edge artifacts.

Image quilting can be used to remove adversarial perturbations by constructing a patch database that only contains patches from "clean" images (without adversarial perturbations); the patches used to create the synthesized image are selected by finding the $K$ nearest neighbors (in pixel space) of the corresponding patch from the adversarial image in the patch database, and picking one of these neighbors uniformly at random. The motivation for this defense is that the resulting image only consists of pixels that were not modified by the adversary — the database of real patches is unlikely to contain the structures that appear in adversarial images.

The right-most column of Figure 2 illustrates the effect of image quilting on adversarial images. Whilst interpretation of these images is more complicated due to the quantization errors that image quilting introduces, it is interesting to note that the absolute differences between quilted original and the quilted adversarial image appear to be smaller in non-homogeneous regions of the image. This suggests that TV minimization and image quilting lead to inherently different defenses.

## 5 EXPERIMENTS

We performed five experiments to test the efficacy of our defenses. The experiment in Section 5.2 considers gray-box attacks: it applies the defenses on adversarial images before using them as input into a convolutional network trained to classify "clean" images. In this setting, the adversary has access to the model architecture and parameters but is unaware of the defense strategy. The experiment in Section 5.3 focuses on a black-box setting: it replaces the convolutional network by networks that were trained on images with a particular input-transformation. The experiment in Section 5.4 combines our defenses with ensembling and model transfer. The experiment in Section 5.5 investigates to what extent networks trained on image-transformations can be attacked in a gray-box setting. The experiment in Section 5.6 compares our defenses with prior work. The setup of our gray-box and black-box experiments is illustrated in Figure 3. Code to reproduce our results is available at https://github.com/facebookresearch/adversarial_image_defenses.

### 5.1 EXPERIMENTAL SETUP

We performed experiments on the ImageNet image classification dataset. The dataset comprises $1.2$ million training images and $50,000$ test images that correspond to one of $1,000$ classes. Our adversarial images are produced by attacking a ResNet-50 model (He et al., 2016). We evaluate our defense strategies against the four adversarial attacks presented in Section 3. We measure the strength of an adversary in terms of its normalized $L_2$-dissimilarity and report classification accu-

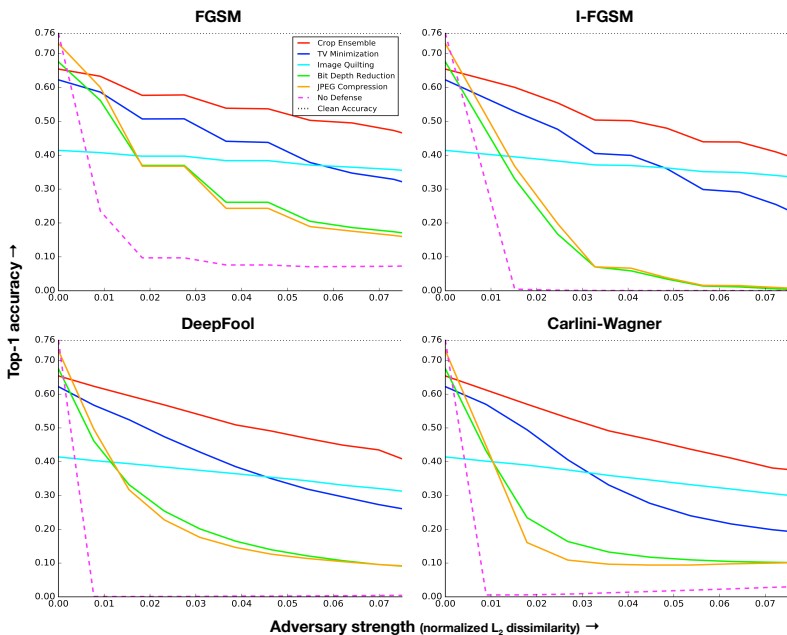

Figure 4: Top-1 classification accuracy of ResNet-50 *tested* on transformed adversarial images produced by four attacks using five image transformations *in a gray-box setting*: (1) cropping-rescaling, (2) bit-depth reduction, (3) JPEG compression, (4) total variance minimization, and (5) image quilting. The dotted line shows the top-1 accuracy of the ResNet-50 model on non-adversarial images, providing an upper bound on the effectiveness of a defense. An $L_2$-dissimilarity of $0.00$ corresponds to the classification accuracy on non-adversarial images. Higher is better.

racies as a function of the normalized $L_2$-dissimilarity. To produce adversarial images like those in Figure 1, we set the normalized $L_2$-dissimilarity for each of the attacks as follows:

- *FGSM.* Increasing the step size $\epsilon$ increases the normalized $L_2$-dissimilarity.
- *I-FGSM.* We fix $M = 10$, and increase $\epsilon$ to increase the normalized $L_2$-dissimilarity.
- *DeepFool.* We fix $M = 5$, and increase $\epsilon$ to increase the normalized $L_2$-dissimilarity.
- *CW-L2.* We fix $\kappa = 0$ and $\lambda_f = 10$, and multiply the resulting perturbation by an appropriately chosen $\epsilon \geq 1$ to alter the normalized $L_2$-dissimilarity.

We fixed the hyperparameters of our defenses in all experiments: specifically, we set pixel dropout probability $p = 0.5$ and the regularization parameter of the total variation minimizer $\lambda_{\text{TV}} = 0.03$. We use a quilting patch size of $5 \times 5$ and a database of $1,000,000$ patches that were randomly selected from the ImageNet training set. We use the nearest neighbor patch (*i.e.*, $K = 1$) for experiments in Sections 5.2 and 5.3, and randomly select a patch from one of $K = 10$ nearest neighbors in all other experiments. In the cropping defense, we sample 30 crops of size $90 \times 90$ from the $224 \times 224$ input image, rescale the crops to $224 \times 224$, and average the model predictions over all crops.

## 5.2 GRAY BOX: IMAGE TRANSFORMATIONS AT TEST TIME

Figure 4 shows the top-1 accuracy of a ResNet-50 tested on transformed adversarial images as a function of the adversary strength for each of the four attacks. Each plot shows results for five different transformations we apply to the images at test time (*viz.*, image cropping-rescaling, bit-depth reduction, JPEG compression, total variation minimization, and image quilting). The dotted line shows the classification error of the ResNet-50 model on images that are not adversarially perturbed, *i.e.*, it gives an upper bound on the accuracy that defenses can achieve.

In line with the results reported in the literature, the four adversaries successfully attack the ResNet-50 model in nearly all cases (FGSM has a slightly lower favorable attack rate of $80-90\%$) when the

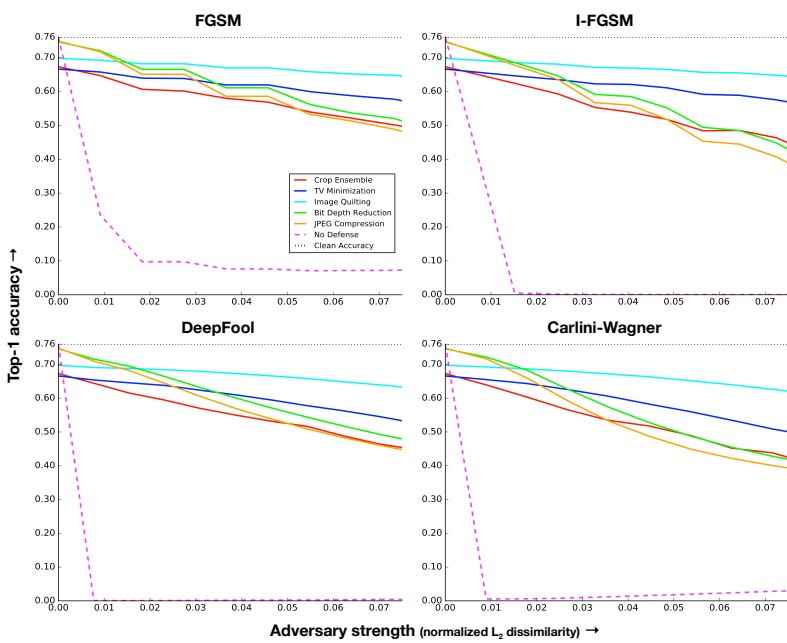

Figure 5: Top-1 classification accuracy of ResNet-50 *trained and tested* on transformed adversarial images produced by four attacks using five image transformations *in a black-box setting*: (1) cropping-rescaling, (2) bit-depth reduction, (3) JPEG compression, (4) total variance minimization, and (5) image quilting. The dotted line represents the top-1 accuracy of the ResNet-50 model on non-adversarial images, providing an upper bound on the effectiveness of a defense. An $L_2$-dissimilarity of $0.00$ corresponds to the classification accuracy on non-adversarial images. Higher is better.

input images are not transformed. The results also show that the proposed image transformations are capable of partly eliminating the effect of the attacks. In particular, ensembling 30 predictions over different, random image crops is very efficient: these predictions are correct for $40-60\%$ of the images (note that $76\%$ is the highest accuracy that one can expect to achieve). This result suggests that adversarial examples are susceptible to changes in the location and scale of the adversarial perturbations. While not as effective, image transformations based on total variation minimization and image quilting also successfully defend against adversarial examples from all four attacks: applying these transformations allows us to classify $30-40\%$ of the images correctly. This result suggests that total variation minimization and image quilting can successfully remove part of the perturbations from adversarial images. In particular, the accuracy of the image-quilting defense hardly deteriorates as the strength of the adversary increases. However, the quilting transformation does severely impact the model's accuracy on non-adversarial images.

## 5.3 BLACK BOX: IMAGE TRANSFORMATIONS AT TRAINING AND TEST TIME

The high relative performance of image cropping-rescaling in 5.2 may be partly explained by the fact that the convolutional network was trained on randomly cropped-rescaled images[2], but not on any of the other transformations. This implies that independent of whether an image is adversarial or not, the network is more robust to image cropping-rescaling than it is to those transformations. The results in Figure 4 suggest that this negatively affects the effectiveness of these defenses, even if the defenses are successful in removing the adversarial perturbation. To investigate this, we trained ResNet-50 models on transformed ImageNet training images. We adopt the standard data augmentation from He et al. (2016), but apply bit-depth reduction, JPEG compression, TV minimization, or image quilting on the resized image crop before feeding it to the network. We measure the classification accuracy of the resulting networks on the same adversarial images as before. Note that this implies that we assume a black-box setting in this experiment.

---

[2]We trained the ResNet-50 model using the data-augmentation scheme of He et al. (2016).

| | Quilting | | | | TVM + Quilting | | | | Cropping + TVM + Quilting | | | |
|---|---|---|---|---|---|---|---|---|---|---|---|---|
| | **RN50** | **RN101** | **DN169** | **Iv4** | **RN50** | **RN101** | **DN169** | **Iv4** | **RN50** | **RN101** | **DN169** | **Iv4** |
| **No Attack** | 70.07 | 72.56 | 70.18 | 73.01 | 72.38 | 74.74 | 73.10 | **75.55** | 72.14 | 74.53 | 72.92 | 75.10 |
| **FGSM** | 65.45 | 68.50 | 65.96 | 67.53 | 65.70 | 68.77 | 67.09 | 69.19 | 66.65 | 69.75 | 67.86 | **70.37** |
| **I-FGSM** | 65.59 | 68.72 | 66.16 | 69.29 | 65.84 | 69.10 | 67.32 | 71.05 | 67.03 | 70.14 | 68.20 | **71.52** |
| **DeepFool** | 65.20 | 68.73 | 65.86 | 68.70 | 65.80 | 69.34 | 67.40 | 71.03 | 67.11 | 70.49 | 68.62 | **71.47** |
| **CW-L2** | 64.11 | 67.72 | 65.00 | 68.14 | 63.99 | 68.20 | 66.08 | 70.13 | 65.31 | 69.14 | 66.96 | **70.50** |

Table 1: Top-1 classification accuracy of ensemble and model transfer defenses (columns) against four black-box attacks (rows). The four networks we use to classify images are ResNet-50 (RN50), ResNet-101 (RN101), DenseNet-169 (DN169), and Inception-v4 (Iv4). Adversarial images are generated by running attacks against the ResNet-50 model, aiming for an average normalized $L_2$-dissimilarity of 0.06. Higher is better. The best defense against each attack is typeset in boldface.

We present the results of these experiments in Figure 5. Training convolutional networks on images that are transformed in the same way as at test time, indeed, dramatically improves the effectiveness of all transformation defenses. In our experiments, the image-quilting defense is particularly effective against strong attacks: it successfully defends against $80-90\%$ of all four attacks, even when the normalized $L_2$-dissimilarity of the attack approaches 0.08.

### 5.4 BLACK BOX: ENSEMBLING AND MODEL TRANSFER

We evaluate the efficacy of (1) ensembling different defenses and (2) "transferring" attacks to different network architectures (in a black-box setting). Specifically, we measured the accuracy of four networks using ensembles of defenses on adversarial images generated to attack a ResNet-50; the four networks we consider are ResNet-50, ResNet-101, DenseNet-169 (Huang et al., 2017), and Inception-v4 (Szegedy et al., 2017). To ensemble the image quilting and TVM defenses, we average the image-quilting prediction (using a weight of 0.5) with model predictions for 10 different TVM reconstructions (with a weight of 0.05 each), re-sampling the pixels used to measure the reconstruction error each time. To combine cropping with other transformations, we first apply those transformations and average predictions over 10 random crops from the transformed images.

The results of our ensembling experiments are presented in Table 1. The results show that gains of $1-2\%$ in classification accuracy can be achieved by ensembling different defenses, whereas transferring attacks to different convolutional network architectures can lead to an improvement of $2-3\%$. Inception-v4 performs best in our experiments, but this may be partly due to that network having a higher accuracy even in non-adversarial settings. Our best black-box defense achieves an accuracy of about $71\%$ against all four defenses: the attacks deteriorate the accuracy of our best classifier (which combines cropping, TVM, image quilting, and model transfer) by at most $6\%$.

### 5.5 GRAY BOX: IMAGE TRANSFORMATIONS AT TRAINING AND TEST TIME

The previous experiments demonstrated the effectiveness of image transformations against adversarial images, in particular, when convolutional networks are re-trained to be robust to those image transformations. In this experiment, we investigate to what extent the resulting networks can be attacked in a gray-box setting in which the adversary has access to those networks (but does not have access to the input transformations applied at test time). We use the four attack methods to generate novel adversarial images against the transformation-robust networks trained in 5.3, and measure the accuracy of the networks on these novel adversarial images in Figure 6.

The results show that bit-depth reduction and JPEG compression are weak defenses in such a gray-box setting. Whilst their relative ordering varies between attack methods, image cropping and rescaling, total variation minimization, and image quilting are fairly robust defenses in the white-box setting. Specifically, networks using these defenses classify up to $50\%$ of adversarial images correctly.

### 5.6 COMPARISON WITH PRIOR WORK

In our final set of experiments, we compare our defenses with the state-of-the-art *ensemble adversarial training* approach proposed by Tramèr et al. (2017). Ensemble adversarial training fits the parameters of a convolutional network on adversarial examples that were generated to attack an

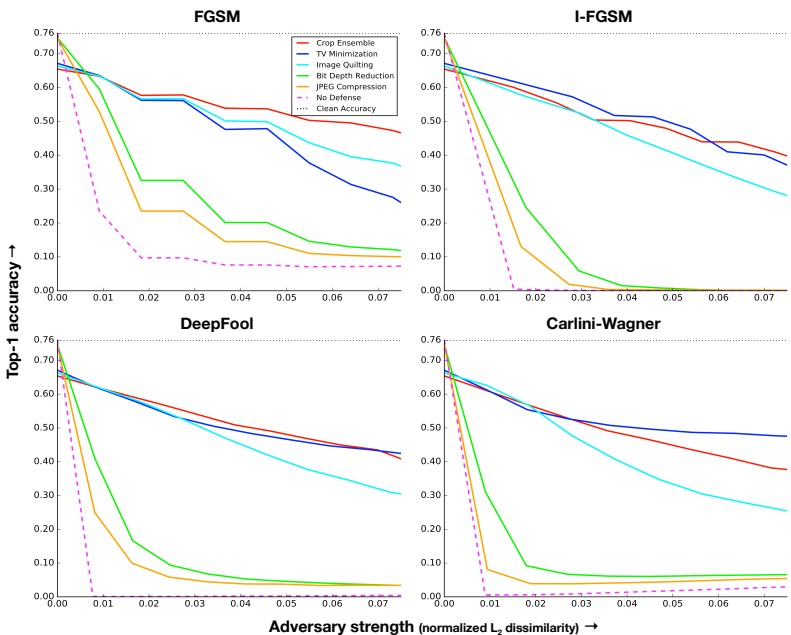

Figure 6: Top-1 classification accuracy of ResNet-50 *trained and tested* on transformed adversarial images produced by four attacks using five image transformations *in a gray-box setting*: (1) cropping-rescaling, (2) bit-depth reduction, (3) JPEG compression, (4) total variance minimization, and (5) image quilting. The dotted line represents the top-1 accuracy of the ResNet-50 model on non-adversarial images, providing an upper bound on the effectiveness of a defense. $L_2$-dissimilarity of 0 corresponds to clean image accuracy. Higher is better.

ensemble of pre-trained models. These adversarial examples are very diverse, which makes the convolutional network being trained robust to a variety of adversarial perturbation. In our experiments, we used the model released by Tramèr et al. (2017): an Inception-Resnet-v2 (Szegedy et al., 2016) trained on adversarial examples generated by FGSM against Inception-Resnet-v2 and Inception-v3 models. We compare the model to our ResNet-50 models with image cropping, total variance minimization, and image quilting defenses. We note that there are two small differences in terms of the assumptions that ensemble adversarial training makes and the assumptions our defenses make: (1) in contrast to ensemble adversarial training, our defenses assume that part of the defense strategy (*viz.*, the input transformation) is unknown to the adversary, and (2) in contrast to ensemble adversarial training, our defenses assume no prior knowledge of the attacks being used. The former difference is advantageous to our defenses, whereas the latter difference gives our defenses a disadvantage compared to ensemble adversarial training.

Table 2 compares the classification accuracies of the defense strategief on adversarial examples with a normalized $L_2$-dissimilarity of 0.06. The results show that ensemble adversarial training works better on FGSM attacks (which it uses at training time), but is outperformed by each of the transformation-based defenses all other attacks. Input transformations particularly outperform ensemble adversarial training against the iterative attacks: our defense are are $18-24\times$ more robust than ensemble adversarial training against DeepFool attacks. Combining cropping, TVM, and quilting increases the accuracy of our defenses against DeepFool gray-box attacks to $51.51\%$ (compared to $1.84\%$ for ensemble adversarial training).

## 6 DISCUSSION

The results from this study suggest there exists a range of image transformations that have the potential to remove adversarial perturbations while preserving the visual content of the image: one merely has to train the convolutional network on images that were transformed in the same way. A critical property that governs which image transformations are most effective *in practice* is whether

|  | Cropping | TVM | Quilting | Ensemble Training (Tramèr et al., 2017) |
|---|---|---|---|---|
| **No Attack** | 65.41 | 66.29 | 69.66 | **80.3** |
| **FGSM** | 49.52 | 31.37 | 39.55 | **69.15** |
| **I-FGSM** | **43.89** | 40.99 | 33.22 | 5.07 |
| **DeepFool** | **44.92** | 44.69 | 34.54 | 1.84 |
| **CW-L2** | 41.06 | **48.41** | 30.51 | 22.23 |

Table 2: Top-1 classification accuracy on images perturbed using attacks against ResNet-50 models trained on input-transformed images, and an Inception-v4 model trained using ensemble adversarial. Adversarial images are generated by running attacks against the models, aiming for an average normalized $L_2$-dissimilarity of 0.06. The best defense against each attack is typeset in boldface.

an adversary can incorporate the transformation in its attack. For instance, median filtering likely is a weak remedy because one can backpropagate through the median filter, which is sufficient to perform any of the attacks described in Section 3. A strong input-transformation defense should, therefore, be non-differentiable and randomized, a strategy has been previously shown to be effective (Wang et al., 2016a;b). Two of our top defenses possess both properties:

1. Both total variation minimization and image quilting are difficult to differentiate through. Specifically, total variation minimization involves solving a complex minimization of a function that is inherently random. Image quilting involves a discrete variable that selects the patch from the database, which is a non-differentiable operation, and the graph-cut optimization complicates the use of differentiable approximations (Maddison et al., 2017).

2. Both total variation minimization and image quilting give rise to *randomized* defenses. Total variation minimization randomly selects the pixels it uses to measure reconstruction error on when creating the denoised image. Image quilting randomly selects one of the $K$ nearest neighbors uniformly at random. The inherent randomness of our defenses makes it difficult to attack the model: it implies the adversary has to find a perturbation that alters the prediction for the entire distribution of images that could be used as input, which is harder than perturbing a single image (Moosavi-Dezfooli et al., 2017).

Our results with gray-box attacks suggest that randomness is particularly important in developing strong defenses. Therefore, we surmise that total variation minimization, image quilting, and related methods (Dong et al., 2011) are stronger defenses than deterministic denoising procedures such as bit-depth reduction, JPEG compression, or non-local means (Buades, 2005). Defenses based on total variation minimization and image quilting also have an advantage over adversarial-training approaches (Kurakin et al., 2016a): an adversarially trained network is differentiable, which implies that it can be attacked using the methods in Section 3. An additional disadvantage of adversarial training is that it focuses on a particular attack; by contrast, transformation-based defenses generalize well across attack methods because they are model-agnostic.

While our study focuses exclusively on image classification, we expect similar defenses to be useful in other domains for which successful attacks have been developed, such as semantic segmentation and speech recognition (Cisse et al., 2017a; Zhang et al., 2017). In speech recognition, for example, total variance minimization can be used to remove perturbations from waveforms, and one could develop "spectrogram quilting" techniques that reconstruct a spectrogram by concatenating "spectrogram patches" along the temporal dimension. We leave such extensions to future work. In future work, we also intend to study combinations of our input-transformation defenses with ensemble adversarial training (Tramèr et al., 2017), and we intend to investigate new attack methods that are specifically designed to circumvent our input-transformation defenses.

## ACKNOWLEDGEMENTS

We thank Kilian Weinberger, Iasonas Kokkinos, Changhan Wang, and the entire Facebook AI Research team for helpful discussions and code support. Chuan Guo is supported in part by NSF grant IIS-1618134.

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
