# OpenReview forum: "Countering Adversarial Images using Input Transformations"
_ICLR.cc/2018/Conference — Accept (Poster)_

### Official Review · AnonReviewer2 · 2017-11-23
**Valuable idea but immature contribution.**

**Rating:** 4
**Confidence:** 3

**Review:**

To increase robustness to adversarial attacks, the paper fundamentally proposes to transform an input image before feeding it to a convolutional network classifier. The purpose of the transformation is to erase the high-frequency signals potentially embedded by an adversarial attack.

Strong points:

* To my knowledge, the proposed defense strategy is novel (even if the idea of transformation has been introduced at https://arxiv.org/abs/1612.01401).

* The writing is reasonably clear (up to the terminology issues discussed among the weak points), and introduces properly the adversarial attacks considered in the work.

* The proposed approach really helps in a black-box scenario (Figure 4). As explained below, the presented investigation is however insufficient to assess whether the proposed defense helps in a true white-box scenario.


Weak points:

* The black-box versus white-box terminology is not appropriate, and confusing. In general, black-box means that the adversary ignores everything from the decision process. Hence, in this case, the adversary does not know about the classification model, nor the defensive method, when used. This corresponds to Figure 3. On the contrary, white-box means that the adversary knows everything about the classification method, including the transformation implemented to make it more robust to attacks. Assimilating the parameters of the transform to a secret key is not correct because those parameters could be inferred by presenting many image samples to the transform and looking at the outcome of the transformation (which is supposed to be available in a 'white-box' paradigm) for those samples.

* Using block diagrams would definitely help in presenting the training/testing and attack/defense schemes investigated in Figure 3, 4, and 5.

* The paper does not discuss the impact of the denfense strategy on the classification performance in absence of adversity.

* The paper lacks of positioning with respect to recent related works, e.g. 'Adversary Resistant Deep Neural Networks with an Application to Malware Detection' in KDD 2017, or 'Building Adversary-Resistant Deep Neural Networks without
Security through Obscurity' at https://arxiv.org/abs/1612.01401.

* In a white-box scenario, the adversary knows about the transformation and the classification model. Hence, an effective and realistic attack should exploit this knowledge. Designing an attack in case of a non differentiable transformation is obviously not trivial since back-propagation can not be used. However, since the proposed transformation primarily aim at removing the high frequency pattern induced by the attack, one could for example design an attack that account for a (linear and differentiable) low-pass filter transformation. Another example of attack that account for transformation knowledge (and would hopefully be more robust than the attacks considered in the manuscript) could be one that alternates between a conventional attack and the transformation.

* If I understand correctly, the classification model considered in Figure 3 has been trained on original images, while the one in Figure 4 has been trained on transformed images. However, in absence of attack, they both achieve 76% accuracy. Is it correct? Does it mean that the transformation does not affect the classification accuracy at all?


Overall, the works investigates an interesting idea, but lacks maturity to be accepted. Therefore, I would only recommend acceptation if room.

Minor issues:

Typo on p7: to change*s*
Clarify poor formulations:
* p1: 'enforce model-specific strategies that enforce model properties such as invariance and smoothness via the learning algorithm or regularization schemes'.
* p1: 'too simple to remove adversarial perturbations from input images sufficiently'

---

> ### Author Response · Authors · 2017-12-22
> **Re: Valuable idea but immature contribution.**
>
> Thank you for your insightful comments on our work, which have been very helpful in improving the paper!
>
> * The black-box versus white-box terminology is not appropriate...
>
> As several public comments have pointed out, the white-box terminology can be misleading. Some of our experiments are performed in a "gray-box" setting in which the adversary has access to the network parameters, but not to the quilting database that acts as a kind of "secret key". We believe that this gray-box setting is of practical interest because the quilting process is stochastic and because the adversary never directly observes the quilted images themselves: this makes it very difficult for the adversary to exactly reproduce the quilted images that the defender produces. Per your suggestion, we have clarified the learning-setting terminology in the revised version of the paper.
>
> * Using block diagrams would definitely help in presenting the training/testing and attack/defense schemes investigated in Figure 3, 4, and 5.
>
> Per your suggestion, we have added block diagrams clarifying the workflow of our attack/defense schemes in the revised version of the paper.
>
> * The paper does not discuss the impact of the defense strategy on the classification performance in absence of adversity.
>
> The first row of Tables 1 and 2 present the accuracy of various defenses on non-adversarial images ("no attack"). In Figures 3, 4 and 5, the y-axis value corresponding to normalized L2-dissimilarity of 0 corresponds to the accuracy on non-adversarial images. We have emphasized this point in the table and figure captions in the revised version of the paper.
>
> * The paper lacks of positioning with respect to recent related works, e.g. 'Adversary Resistant Deep Neural Networks with an Application to Malware Detection' in KDD 2017, or 'Building Adversary-Resistant Deep Neural Networks without Security through Obscurity' at https://arxiv.org/abs/1612.01401.
>
> Thank you for pointing out these references, which we were unaware of at the time of submission. Both approaches are similar to our defenses in the sense that they focus on non-differentiable, stochastic transformations. Having said that, there are also substantial differences between our study and those related works. The first paper relies on LLE to represent data points as a linear combination of nearest neighbors: this approach may certainly be suitable for certain kinds of data, but is unlikely to work very well in extremely high-dimensional spaces such as the ImageNet pixel space. The second paper's approach of randomly removing blocks of pixels is related to our image-cropping baseline defense, which is one of our baselines. We have included positioning with respect to these works in the revised version of the paper.
>
> * In a white-box scenario, the adversary knows about the transformation and the classification model. Hence, an effective and realistic attack should exploit this knowledge...
>
> In white-box settings, it may, indeed, be possible to devise attacks that are tailored towards a particular defense. In our work, we have tried to make the development of such attacks non-trivial by making our defenses non-differentiable and stochastic. Having said that, it may certainly be possible to devise attack strategies that are successful nevertheless (such as the strategy sketched in our response to AnonReviewer3). We leave the investigation of attacks that are tailored to our defenses to future work.
>
> * If I understand correctly, the classification model considered in Figure 3 has been trained on original images, while the one in Figure 4 has been trained on transformed images. However, in absence of attack, they both achieve 76% accuracy. Is it correct? Does it mean that the transformation does not affect the classification accuracy at all?
>
> The 76% accuracy is obtained by a convolutional network that is trained and tested on images on which no defense (i.e., input transformation) is applied. The "no defense" baseline is this exactly the same in both Figures 3 and 4. For defenses such as TV minimization and quilting, the accuracy on non-adversarial images is lower (both in Figure 3 and 4), which shows that the transformations, indeed, do negatively impact classification accuracy on non-adversarial images.

---

### Official Review · AnonReviewer1 · 2017-11-27
**Well argumented, solid work**

**Rating:** 8
**Confidence:** 4

**Review:**

Summary: This works proposes strategies to make neural networks less sensitive to adversarial attacks. They consist into applying different transformations to the images, such as quantization, JPEG compression, total variation minimization and image quilting. Four adversarial attacks strategies are considered to attack a Resnet50 model for classification of Imagenet images.
Experiments are conducted in a black box setting (when the model to attack is unknown by the adversary) or white box setting (the model and defense strategy are known by the adversary).
60% of attacks are countered in this last most difficult setting.
The previous best approach for this task consists in ensemble training and is attack specific. It is therefore pretty robust to the attack it was trained on but is largely outperformed by the authors methods that manage to reduce the classifier error drop below 25%.

Comments: The paper is well written, the proposed methods are well adapted to the task and lead to satisfying results.

The discussion remarks are particularly interesting: the non differentiability of the total variation and image quilting methods seems to be the key to their best performance in practice.
Minor: the bibliography should be uniformed.

---

> ### Author Response · Authors · 2017-12-22
> **Re: Well argumented, solid work**
>
> Thanks for your positive evaluation of our paper! Per your suggestion, we have updated the bibliography entries to make them uniform.

---

### Official Review · AnonReviewer3 · 2017-11-28

**Rating:** 7
**Confidence:** 3

**Review:**

 The paper investigates using input transformation techniques as a defence against adversarial examples. The authors evaluate a number of simple defences that are based on input transformations such TV minimization and image quilting and compare it against previously proposed ideas of JPEG compression and decompression and random crops.  The authors have evaluated their defences against four main kinds of adversarial attacks.

The main takeaways of the paper are to incorporate transformations that are non-differentiable and randomised. Both TV minimisation and image quilting have that property and show good performance in withstanding adversarial attacks in various settings.

One argument that I am not sure would be applicable perhaps and could be used by adversarial attacks is as follows: If the defence uses image quilting for instance and obtains an image $P$ that approximates the original observation $X$, it could be possible to use a model based approach that obtains an observation $Q$ that is close to $P$ which can be attacked using adversarial attacks. Would this observation then be vulnerable to such attacks? This could perhaps be explored in future.

The paper provides useful contributions in forming model agnostic defences that could be further investigated. The authors show that the simple input transformations advocated work against the major kind of attacks. The input transformations of TV minimization and image quilting share varying characteristics in terms of being sensitive to various kinds of attacks and therefore can be combined. The evaluation is carried out on ImageNet dataset with large number of examples.

---

> ### Author Response · Authors · 2017-12-22
> **Re: Review**
>
> Thank you for your insightful comments, and positive evaluation of work!
>
> Regarding model-based approaches for attacking our quilting defense: we agree this may the most viable option for attacking our defense. As you suggest, it may be possible for the adversary to construct its own patch database, and use it to construct quilted images that may be sufficiently similar to the quilted image created using our "secret database". The remaining issue for the adversary is then to backpropagate gradients through the quilting transformation: the adversary may be able to do this by training a pixel-to-pixel network that learns to produce the quilted image given an original image, and using this network to approximate gradients. We intend to investigate such attack approaches in future work. We have updated our paragraph describing future work to reflect this.

---

### Public Comment · (anonymous) · 2017-11-03
**White-box attacks with knowledge of defense transformation**

Interesting paper! It wasn't clear to me whether you evaluated attacks (white-box or black-box) with knowledge of the defensive technique being used? For JPEG or TV-minimization for instance, this would mean back-propagating through the transformation step. Analogously, one could incorporate randomized procedures such as cropping, pixel dropout or quilting into the adversarial example generation procedure, either in a white-box setting or in a black-box setting over a locally trained model with a similar defense.

Some prior works [1,2] seem to indicate that various types of transformations do actually remain vulnerable to adversaries with such tailored attacks. ([1] considered random crops and found that an attack tailored to a particular cropping mechanism was still effective. [2] looked at random pixel dropout and found that computing white-box or black-box attacks for multiple randomly chosen dropout masks generalized well to unseen random masks).

[1] Foveation-based Mechanisms Alleviate Adversarial Examples, https://arxiv.org/abs/1511.06292
[2] Adversarial Examples Are Not Easily Detected: Bypassing Ten Detection Methods, https://arxiv.org/abs/1705.07263

---

> ### Author Response · Authors · 2017-11-03
> **Re: White-box attacks with knowledge of defense transformation**
>
> Thank you for your comment! We were unaware of [1] at the time of submission. We will include it as a citation. Based on our understanding, [2] applies Carlini-Wagner's attack against a target loss that averages the prediction over multiple fixed models with random network weight dropout rather than random pixel dropout, but the two techniques are certainly related.
>
> In regards to attacking the transformation defenses, independent of [1], we have observed that it is possible to produce adversarial examples that are invariant to crop location and scale. By randomly selecting a crop of size 135x135 each iteration to compute the loss function for CW-L2, the resulting adversarial examples can reduce the accuracy of crop defense to 30% at an average L2-dissimilarity of 0.045.
>
> Enhancing the attack with random pixel dropping can, indeed, reduce the effectiveness of TV minimization significantly. Using a pixel dropout mask with drop probability 0.1 each iteration, CW-L2 can reduce the accuracy of TV minimization defense to 9% at an average L2-dissimilarity of 0.06. However, we do not have a good idea of how to backpropagate through the quilting transformation, as the construction is stochastic and non-differentiable in nature. Nevertheless, section 5.5 of our paper does show that it is possible to successfully attack the quilting defense in some cases even without knowledge of this transformation; we presume because the convolutional filters reveal some information about the patch database used.
>
> The attacks we use for the white-box setting do not have knowledge of the defense mechanism used.
>
> We will clarify these points in the paper.

---

> > ### Public Comment · (anonymous) · 2017-11-03
> > **Re: White-box attacks with knowledge of defense transformation**
> >
> > Thanks for these clarifications! It might make sense to introduce a new term to characterize the threat model here, as "white-box" typically refers to full information about the defense. I think "grey-box" has been used before although it isn't very evocative either.

---

> > > ### Author Response · Authors · 2017-11-05
> > > **Re: White-box attacks with knowledge of defense transformation**
> > >
> > > We agree that the terminology of white-box is ambiguous, in particular, in the context of randomized defenses (is the adversary allowed access to the random seed?). In cryptography, Kerckhoffs's principle prescribes the use of a "secret key". One could make the argument that the patch database used by image quilting is an implementation of such a secret key. To the best of our knowledge, there is no consensus yet in the community yet on what a "secret key" may and may not contain in the context of defenses against adversarial examples.
> > >
> > > In our current paper, "white-box" refers to access the model parameters; other parameters (random seed, patch database) are considered to be part of the secret key. We will add a section to our paper clarifying the terminology.

---

### Public Comment · (anonymous) · 2017-11-03
**PGD evaluation missing**

I am curious why the authors have not evaluated their defense methods against the PGD attack of Madry et al https://arxiv.org/abs/1706.06083, which is the strongest first order adversary possible.

---

> ### Author Response · Authors · 2017-11-03
> **Re: PGD evaluation missing**
>
> For our experiments, we selected four attacks that we believe are representative of the large number of attacks that people have proposed. Since PGD is related to I-FGSM (the main difference between the two is the projection step after every iteration), we expect that our defenses will have similar performance against PGD.
>
> We performed a small experiment with PGD to confirm this. We created attacks with an average L2-dissimilarity of 0.06, and find that the accuracy of our defenses against white-box I-FGSM/PGD attacks are: no defense 0%/0%, crop ensemble 44%/48%, TVM 29%/36%, and image quilting 35%/37%. These results suggest that the effectiveness of our defenses against PGD is similar to their effectiveness against I-FGSM. We will add these results to the paper.

---

> > ### Public Comment · (anonymous) · 2017-11-04
> > **Three comments**
> >
> > 1) Projecting after every step would make a lot of difference if the number of iterations and step length is high, since typically gradient signal from outside the epsilon ball is not that useful in finding adversarial examples.
> >
> > 2) I am wondering what kind of accuracy your defense would get on MNIST/CIFAR-10, so one could compare to vanilla adversarial training as in Madry et al. It is not clear at the moment whether it would improve on adversarial training with PGD.
> >
> > 3) Since some of the transformations are non-differentiable you could also have settings where the attacker knows your transformations and attack your transformed inputs (rather than the original image) - as the other commenter says, what you study is the "grey-box" case and not truly the "white-box" case. I would expect that in the truly "white-box case", it is no more robust against such attacks than just adversarial training with PGD.

---

> > > ### Author Response · Authors · 2017-11-05
> > > **Re: Three comments**
> > >
> > > 1) We have performed experiments with our defenses against PGD; see our previous comment for the results of those experiments, which we will add to the paper. Our results do not suggest substantial differences in the effectiveness of our defenses between PGD and I-FGSM.
> > >
> > > 2) Our proposed defenses are not intended to compete with adversarial-training-based defenses: the two defenses can be used together and are likely to be complementary. We chose ImageNet to conduct our experiment for the following two reasons:
> > >
> > > - The interest in adversarial examples mainly stems from the concern of use of computer vision models in real-world applications such as self-driving cars and image-classification services. In these settings, the input to the model has high resolution and diverse content; ImageNet more closely resembles this scenario than MNIST or CIFAR.
> > >
> > > - Defending a model that performs classification on ImageNet is inherently more difficult than defending a MNIST or CIFAR classification model, since the model must output very diverse class labels and the model's prediction is often uncertain. Moreover, the input dimensionality for ImageNet is much higher (~150000 compared to 768 for MNIST and 3072 for CIFAR-10), which gives the attacker much more maneuverability.
> > >
> > > 3) We agree that the terminology of white-box is ambiguous, in particular, in the context of randomized defenses (is the adversary allowed access to the random seed?). In cryptography, Kerckhoffs's principle prescribes the use of a "secret key". One could make the argument that the patch database used by image quilting is an implementation of such a secret key. To the best of our knowledge, there is no consensus yet in the community yet on what a "secret key" may and may not contain in the context of defenses against adversarial examples. In our current paper, "white-box" refers to access the model parameters; other parameters (random seed, patch database) are considered to be part of the secret key. We will add a section to our paper clarifying the terminology.

---

### Public Comment · (anonymous) · 2017-11-09
**Performance on clean examples**

The proposed several transformation methods are quite effective in recognizing the adversarial examples. For most existing works, people are trying to increase the performance on adversarial examples while maintaining the accuracy of clean examples. However, in table 2, I found there is a huge drop in accuracy for all your methods on clean examples.
I wonder can you solve this problem by tuning some hyper-parameters to balance the performance of no attack and with attack?

---

> ### Author Response · Authors · 2017-11-11
> **Re: Performance on clean examples**
>
> The transformations we studied, indeed, all have some hyper-parameter that controls how lossy the transformation is, and can be used to trade off clean accuracy and adversarial accuracy. These hyper-parameters are: crop ratio for the crop-rescale transform, pixel drop rate and regularization parameter for total variation minimization, and patch size for image quilting. For instance, using a larger patch size in image quilting will remove more of the adversarial perturbation (which likely leads to higher adversarial accuracy), but also affects clean images which deteriorates clean accuracy.
>
> We selected hyper-parameter that achieve high adversarial accuracy, but one may choose to set them differently depending on the user's needs. We surmise it may be possible to achieve high clear and adversarial accuracy by ensembling predictions over multiple hyper-parameter settings, but further experimentation is needed to confirm this hypothesis.

---

> ### Public Comment · (anonymous) · 2017-11-16
> **The "huge drop"**
>
> The authors' result is not surprising at all. It is actually one of the few works on the defense side which makes sense to me.
> Personally, I have not seen any strong evidence suggesting that we can generally maintain a high accuracy on clean and adversarial samples simultaneously. We have to take any such idealistic results on ImageNet with a grain of salt. In most of these works, either the defense is super weak (i.e., easily attackable) or something is hidden under the carpet.

---

> > ### Public Comment · (anonymous) · 2017-11-17
> > **Re: The "huge drop"**
> >
> > You might want to check out https://openreview.net/forum?id=S18Su--CW where the authors show that simple quantization (depth reduction) leads to a loss of accuracy on clean examples, but you can get around it by discretizing your input. Also simple quantization maintains a roughly linear response of the classifier to it's input (see Figure 1), which is hypothesized as the cause of non-robustness to adversarial attacks (Goodfellow, 2014). The attacks proposed are also not "grey-box" like in this paper, i.e. the attacker has knowledge of the transformations and can attack using "discretized" versions of various iterated algorithms. The defense of this paper is essentially defense by "obfuscation", under the assumption that the attacker does not know what "obfuscation" has happened.

---

> > > ### Public Comment · (anonymous) · 2017-11-29
> > > **Re: Re: The "huge drop"**
> > >
> > > This work considers ImageNet which is a far more challenging dataset. After reading the other submission (which is quite interesting too), I humbly think that this work is the most sensible ICLR submission on defending against adversarial attacks.

---

### Public Comment · (anonymous) · 2018-01-10
**Parameter settings used in experiments**

I'm trying to reproduce the experimental results shown in this paper. Thank you for giving a detailed description of parameter settings in Section 5.1 -- this makes it a lot more straightforward.

One thing that I noticed was missing was the tile overlap for the image quilting defense. You mention that you use 5x5 tiles, with a database of 1M tiles, and randomly select from the 10 nearest neighbors. In Section 4.3 and Section 6, you mention using minimum graph cuts in boundary regions, but this only applies when you have overlapping tiles. What was the overlap parameter used in the experiments?

---

> ### Author Response · Authors · 2018-01-10
> **The overlap size is 2 pixels**
>
> Apologies for leaving out this detail! We used an overlap of 2 pixels between the 5x5 patches. We will update the paper with this information.

---

> > ### Public Comment · (anonymous) · 2018-01-10
> > **Thanks**
> >
> > Wow, that was a quick response. Thanks!

---

### Public Comment · (anonymous) · 2018-01-12
**Unreasonable threat model assumptions**

It appears that in the revised manuscript, the authors completely change their threat model. In the new draft of the paper, the authors add the sentence "our defenses assume that part of the defense strategy (viz., the input transformation) is unknown to the adversary".

This is a completely unreasonable assumption. Any algorithm which hopes to be secure must allow the adversary to, at the very least, understand what the defense is that's being used. Consider a world where the defense here is implemented in practice: any attacker in the world could just go look up the paper, read the description of the algorithm, and know how it works.

The authors mention Kerckhoffs's Principle in a comment below, and that's exactly what is being violated here. It is perfectly reasonable to assume the adversary does not have the model parameters or training data. But declaring the entire defense process a secret is exactly what Kerckhoffs's Principle says is *not* okay.

The current threat model of this paper appears to be arguing for security through obscurity, and that is not a reasonable defense.

---

> ### Author Response · Authors · 2018-01-12
> **Re: Unreasonable threat model assumptions**
>
> We agree that security through obscurity is inherently weak and ineffective. However, the defenses that we evaluated, in particular TV minimization and image quilting, are stochastic in nature. This allows the defense to randomize its transformation, and hence prevents the adversary from knowing the exact transformation being applied even if the defense strategy is known. The image quilting defense enjoys the additional property that the patch database used to construct the quilted images can be considered as the secret key, which obeys Kerckhoff's principle: even if the defense strategy is known, the adversary cannot apply the same quilting transformation if he does not have access to the patch database.

---

> > ### Public Comment · (anonymous) · 2018-01-12
> > **Thank you for the clarification**
> >
> > I believe that clarifies the statement made. The claim being made is not that the adversary does not have any knowledge of the defense, but just that the adversary doesn't know the exact choices of randomness that the defender makes.

---

> > > ### Public Comment · (anonymous) · 2018-01-28
> > > **Randomness does not constitute strong defense**
> > >
> > > There is a paper “Adversarial Examples Are Not Easily Detected: Bypassing Ten Detection Methods” (https://nicholas.carlini.com/papers/2017_aisec_breakingdetection.pdf) which explores how stochastic model could be de-randomized and successfully attacked.
> > >
> > > Thus while randomness makes an attack harder, it does not prevent white-box adversary from conducting successful attack.

---

### Public Comment · (anonymous) · 2018-01-26
**Previous Work**

Additional prior works in this area [1] have shown transformations to be effective in countering adversarial examples, including cropping and resizing.

As I did not see the paper cited, I thought it might be worth bringing them to the author's attention, especially given the similar nature of the work.

[1] https://arxiv.org/abs/1610.04256

---

> ### Author Response · Authors · 2018-01-26
> **Thanks**
>
> Thanks for pointing us to this work! We will update our paper to refer to it.

---

### Public Comment · (anonymous) · 2018-01-28
**Problems in methodology**

There are few problems in methodology which can make the results appear to be overly optimistic.

The first problem is the way the authors have chosen the size of their adversarial perturbation.
They claim (section 2 of the paper) that “success rate is generally measured as a function of the magnitude of the perturbations performed by the attack, using the normalized L2-dissimilarity” and provide formula for L2 dissimilarity in equation (1).
This claim is not supported by any references. Moreover, many papers which the authors site (including C&W, FGSM and I-FGSM attacks or any papers about adversarial training) actually use different metrics to measure strength of an adversary. It could be either L2-distance to adversarial example (like in C&W attack) or accuracy of the defense against adversarial examples with fixed L_{\infty} size of perturbation (like in FGSM and I-FGSM attacks). The authors’ L2 dissimilarity metric makes it impossible to infer corresponding L_2 or L_{\infty} size of adversarial perturbation, and thus makes it almost impossible to fairly compare results of the paper to any existing work.

The second problem is the fact that the authors seem to misunderstand the I-FGSM method. As initially described in https://arxiv.org/abs/1607.02533, I-FGSM does clipping after each step (which is called “projection” in PGD). However, according to eq (3) from the paper, the authors do not use projection or clipping. Without projection after each step, the attack may be much weaker than it should be because it will end up greatly overshooting its final epsilon boundary (and have one large clipping at the end).

Also, the authors say "Since PGD is related to I-FGSM (the main difference between the two is the projection step after every iteration), we expect that our defenses will have similar performance against PGD." Actually, the most important difference between PGD and I-FGSM is not the projection step (both attacks have it), but the fact that PGD starts with random noise and also performs random restarts throughout the attack -- both of which makes the attack much stronger. Because of this, it appears that the PGD accuracies from the comments may be incorrect.

The last problem is the number of steps used in iterative attacks. The authors use 10 steps for I-FGSM attack and 5 steps for DeepFool. This might have been okay if their step size per iteration was large, but in the paper, the authors say their step size was just large enough to achieve their desired L2-dissimilarity. However to achieve a strong enough attack, step_size_per_iteration * number_of_steps should be greater than the final epsilon boundary.
Ideally authors need to explore how number of iterations affect strength of the adversary, assuming fixed step size. An example of good justification for choosing number of iterations is figure 1 from https://arxiv.org/abs/1706.06083, which clearly shows that at least 40 - 50 iterations are needed to achieve strong attack in their case.

---

### Public Comment · (anonymous) · 2018-01-28
**Confusing statements about white-box attack**

All the attacks tested in the paper are done in either “gray-box” or “black-box” settings. In both cases, the adversary does not have full knowledge about the defense being used.
However, the authors’ replies to the reviewers’ comments as well as statements in the paper (e.g. “we focus on increasing the effectiveness of model-agnostic defense strategies by developing approaches that ... are still effective in settings in which the adversary has information on the defense strategy being used”) create an impression that the authors claim their defenses would work in true “white box” case.

It is worth noting that in the true white box case, the adversary knows both the model and all preprocessing techniques. The authors do not study this case, and their claim of effectiveness is based on the fact that their proposed preprocessing techniques (TV and image quilting) are stochastic and non-differentiable.

It was already mentioned in the other comments that non-differentiable transformations make it hard to use common gradient based attacks. As a note, it is still possible by training a substitute neural network to approximate the non-differentiable transformation.

Moreover an attack on stochastic defense was successfully performed in “Adversarial Examples Are Not Easily Detected: Bypassing Ten Detection Methods” (https://nicholas.carlini.com/papers/2017_aisec_breakingdetection.pdf).
The general idea is the following. Suppose you have a stochastic model f which maps the input into the logits vector (in this case, f will include both input transformation and classifier). Then, you can de-randomize f by sampling multiple instance f_1, ... , f_N of this model. Then you can attack an ensemble of deterministic functions f_1, …, f_N and find adversarial examples that fool all (or most) of them. If the functions f_i happen to be non-differentiable, then you can approximate them with differentiable functions and then conduct an attack.

Given that the white box attack against stochastic defenses proposed by Carlini et al is well known, we think it was important for the authors to evaluate their defense against this attack. The authors seem to imply this is not an immediate concern (e.g. “"We leave the investigation of attacks that are tailored to our defenses to future work" in response to Reviewer #2), but we think that any paper that proposes a stochastic defense should be evaluated against this attack to see how well it performs in the white box case.

---

> ### Public Comment · (anonymous) · 2018-01-28
> **Security through obscurity and publishing potential defenses**
>
> Due to the concerns described above, it appears that the authors’ defense unfortunately reduces to security through obscurity. When this concern was brought up in the comments, the authors claim that this is not the case because while the adversary may know the defense strategy, they will not know the exact randomized transformation. However, as described above, this is not true, and the authors don’t properly address the case when adversary actually knows the defenses.
> The problem here is that as soon as the paper is published, any potential adversary knows that these methods could be used as a defense -- and thus they can adjust their attacks accordingly.

---

### Decision · Program_Chairs · 2018-01-29
**ICLR 2018 Conference Acceptance Decision**

**Decision:**

Accept (Poster)

**Comment:**

A well written paper proposing some reasonable approaches to counter adversarial images. Proposed approaches include non-differentiable and randomized methods. Anonymous commentators pushed upon and cleared up some important issues regarding white, black and gray "box" settings. The approach appears to be a plausible defence strategy. One reviewers is a hold out on acceptance, but is open to the idea. The authors responded to the points of this reviewer sufficiently. The AC recommends accept.